# Circumferential Material Flow in the Hydroforming of Overlapping Blanks

**Cong Han [1,2,*] and Hao Feng [1]**

[1] School of Materials Science and Engineering, Harbin Institute of Technology, Harbin 150001, China; kingmouseboca@126.com

[2] National Key Laboratory of Precision Hot Processing of Metals, Harbin Institute of Technology, Harbin 150001, China

\* Correspondence: conghan@hit.edu.cn; Tel.: +86-451-8641-7917

**Abstract:** The hydroforming of the overlapping blanks is a forming process where overlapping tubular blanks are used instead of tubes to enhance the forming limit and improve the thickness distribution. A distinguishing characteristic of the hydroforming of overlapping blanks is that the material can flow along the circumferential direction easily. In this research, the circumferential material flow was investigated using overlapping blanks with axial constraints to study the circumferential material flow in the hydroforming of a variable-diameter part. AISI 304 stainless steel blanks were selected for numerical simulation and experimental research. The circumferential material flow distribution was obtained from the profile at the edge of the overlap. The peak value located at the middle cross-section. In addition, the circumferential material flow could be also reflected in the variation of the overlap angle. The variation of the overlap angle kept increasing as the initial overlap angle increased but the improvement of the thickness distribution did not. There was an optimal initial overlap angle to minimize the thinning ratio. An optimal thickness distribution was obtained when the initial angle was 120° for the hydroforming of the variable-diameter part with an expansion of 31.6%.

**Keywords:** hydroforming; overlapping blank; variable-diameter part; thickness

## 1. Introduction

In the past decades, the demand for lightweight structures and components has been increasing with the rapid development of the automotive industry [1]. A variety of tubular components, i.e., exhaust pipes, chassis systems and structural components are designed and assembled in automobiles. The manufacturing processes of these components should be straightforward, efficient and reliable [2]. Tube hydroforming has proved itself as an advanced metal forming process to manufacture these components because of its part consolidation, weight reduction and lower costs [3,4]. Variable-diameter components, as a kind of typical hydroformed parts, are widely used in the exhaust pipes of automotives. The diameter of the middle region of the variable-diameter tubular part is much larger than the end [5].

With regard to the tube hydroforming process of variable-diameter tubular components, there are certain limitations on the degree of deformation owing to the hydroformability of the material used [6]. Up to now, a number of studies on the hydroformability of tubular components have been reported [7]. A high strain hardening exponent and a high plastic anisotropy of the material were effective to improve the wall thickness [8,9]. Lubrication conditions had a significant effect on defects of hydroformed parts, such as wrinkling, buckling and cracking [10,11]. The thickness distribution was relatively uniform due to the low friction coefficient when an appropriate lubricant was used [12]. The loading path (the relationship of the internal pressure and the axial feeding) had a significant effect on the formability for a certain material [13]. On the basis of it, the useful wrinkles have been proposed

and applied to enlarge the process window [14]. It has been demonstrated that useful wrinkles can improve the forming limit and thickness distribution by increasing axial feeding [15,16]. Wrinkling defects, however, may occur easily due to the axial feeding if the diameter-to-thickness ratio (*D/t*) of the part is extremely large [17]. It is not feasible to enhance the forming limit by increasing the axial feeding in this situation.

Therefore, a novel approach was proposed, in which overlapping tubular blanks were used instead of closed cross-section tubes [18]. A sound spherical part with an expansion of 60.0% was obtained by using this approach, whereas the maximum expansion was only 46.1% using tubes with closed cross-sections. The key to the enhancement of forming limits was the material flow along the circumferential direction. However, the material flowed along both axial and circumferential directions simultaneously due to the free ends of the overlapping blank. In order to study the material flow along the circumferential direction, the influence of the axial direction should be avoided during the hydroforming of overlapping blanks.

In this paper, the circumferential material flow was investigated using overlapping blanks with axial constraints in the hydroforming of a variable-diameter part [19]. The shape of the variable-diameter part was cylindrical to guarantee that the expansion rates were equal at the bulging area. AISI 304 stainless steel blanks were selected for numerical simulation and experimental research. The circumferential material flow distribution was reflected in the profile of the blank edge. The effect of the overlap angle on the variation of the overlap and thickness distribution was studied.

## 2. Materials and Methods

### 2.1. Samples and Material

The sample was a variable-diameter part, as shown in Figure 1. The maximum diameter was 100 mm at the middle region and the diameter was 76 mm at each end. The diameter of the middle region was 31.6% larger than the end. The length of the middle region was 56 mm. The angle of the transition area was 20°. The radius of the fillet was 15 mm in the transitional area.

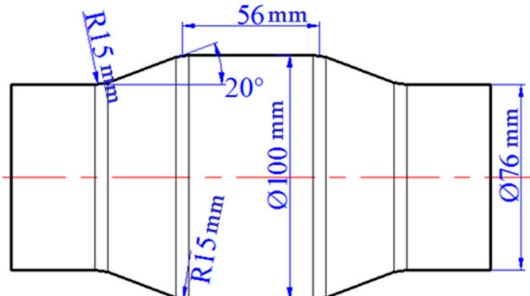

**Figure 1.** Shape and dimension of the sample.

Overlapping tubular blanks were obtained from sheet blanks after bending and used for the hydroforming of the variable-diameter part. In comparison with the closed cross-sectional tube which was shown in Figure 2a, the cross-section of the overlapping blank was open. The overlap consisted of the inner and the outer layers, as shown in Figure 2b.

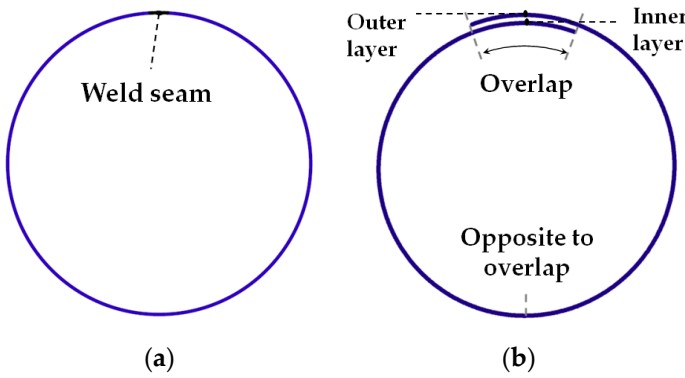

**Figure 2.** Shape of cross-sections: (**a**) tube blank; (**b**) overlapping blank.

Figure 3a shows the shape and dimension of the overlapping blank. The outer diameter and the length of the tubular blank were 76 and 250 mm, respectively. The level of the overlap was determined by the initial overlap angle $\alpha$ that was defined as the angle between the edges of the inner layer and the outer layer.

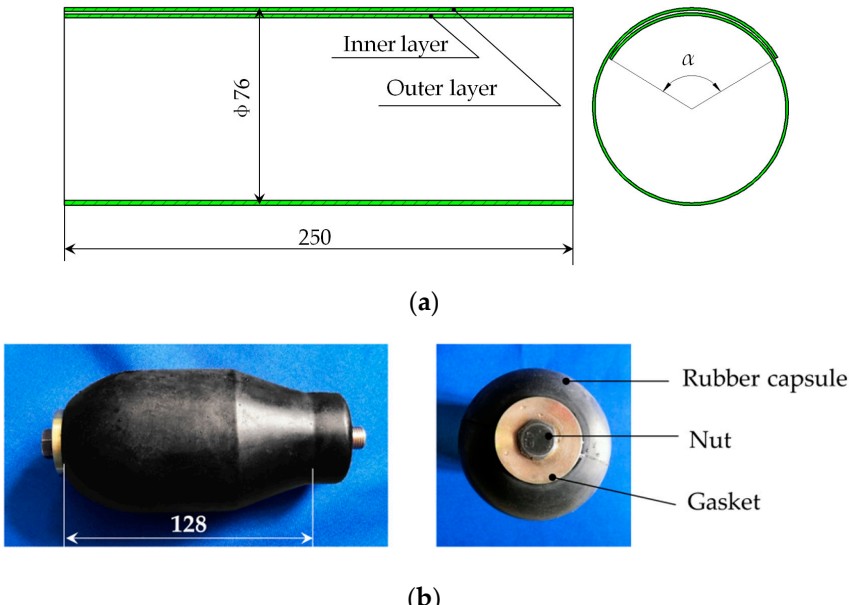

**Figure 3.** Shape and dimension of: (**a**) the overlapping blank; (**b**) the elastic body.

An elastic body was used for the expansion of the overlapping blank. The function of the elastic body was achieved by rubber capsules that were available on the market. The diameter of the rubber capsule ranged from 50 to 72 mm. The total length of the rubber was 150 mm and the length of the bulging area of the rubber was 128 mm, as shown in Figure 3b. The maximum elastic elongation of the rubber was 100% at least, which was much higher than the expansion rate of the blank. Therefore, the rubber capsule was capable of the hydroforming of the variable-diameter part. There was an entrance for the liquid medium at the right of the elastic body. At the left, the gaskets and the nut could seal the left end of the rubber capsule.

AISI 304 stainless steel blanks were used in the simulation and experimental research. The initial wall thickness of the blank was 0.5 mm. The uniaxial tensile tests were conducted on an AGX-plus 20KN/5KN (SHIMADZU Corp., Kyoto, Japan) machine along the rolling direction of the sheet blanks. Common tensile properties like yield and ultimate tensile strength were obtained from the true stress-strain curve, as shown in Figure 4. The mechanical properties are given in Table 1.

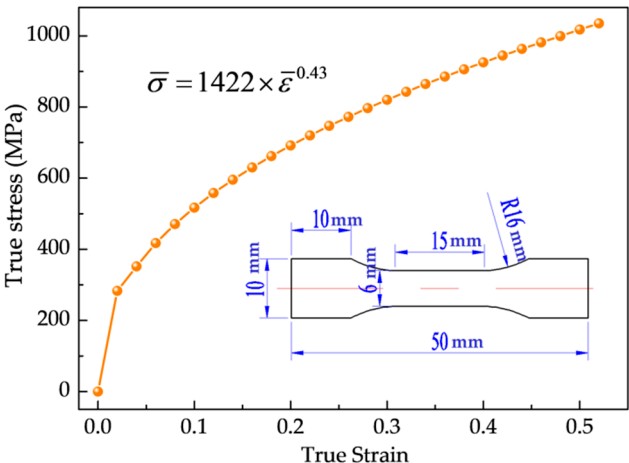

**Figure 4.** Flow stress curve of AISI 304 stainless steel.

**Table 1.** Mechanical properties of AISI 304 stainless steel.

| Elastic Modulus $E$ (GPa) | Yield Strength $\sigma_s$ (MPa) | Ultimate Strength $\sigma_b$ (MPa) | Elongation $\delta$ (%) |
|:---:|:---:|:---:|:---:|
| 208 | 287 | 903 | 52.6 |

### 2.2. Experimental Setup

The experiments were conducted on the 10 MN hydroforming machine. Figure 5 shows the schematic diagram of the experimental setup for the hydroforming of the variable-diameter part. The hydroforming tools mainly consisted of a lower die, an upper die, two blocks and a self-sealing elastic body.

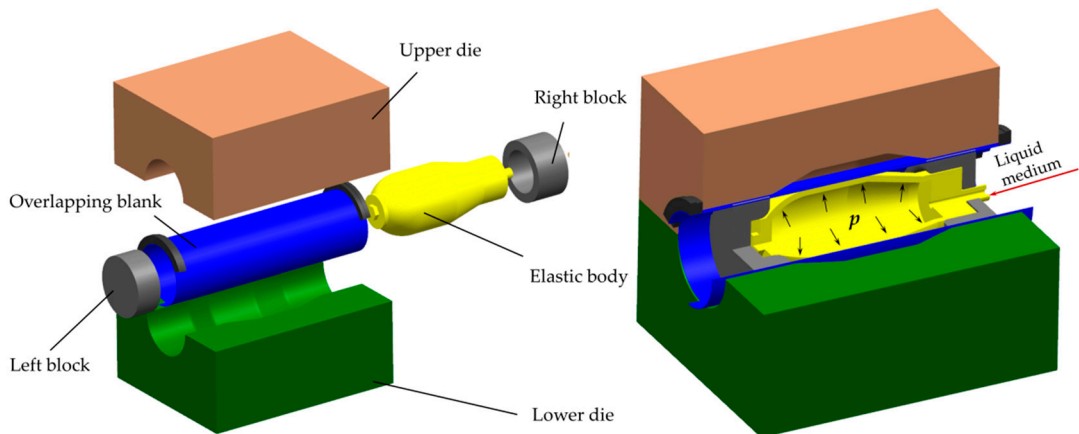

**Figure 5.** Schematic diagram of the experimental setup.

The liquid medium flowed into the rubber capsule cavity via the right entrance of the elastic body during the hydroforming process. The two blocks were placed against the ends of the elastic body to prevent it from extending along the axial direction, as shown in Figure 6. The pressurized liquid was stored in the expansive elastic body which pushed the overlapping blank to bulge as the internal pressure increased. When the process finished, the internal pressure was unloaded and the capsule could be taken out from the blank due to its high elasticity property.

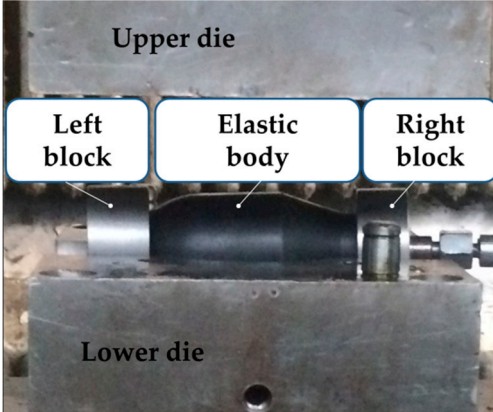

**Figure 6.** The self-sealing elastic body used for the experiment.

### 2.3. Finite Element Model

The numerical simulation was conducted with FEM software Abaqus 6.13. (SIMULIA Corp., Providence, RI, USA) Figure 7 shows the finite element model used for the hydroforming of a variable-diameter part with overlapping tubular blanks. The die and the blocks were defined as a rigid body. The overlapping blank was defined as a deformed body using the elastoplastic model. The elastic body was defined as a hyperelastic model and its original shape was simplified into a cylinder in the finite element model. The influence of the elastic body shape was limited to the simulation results due to the high elasticity.

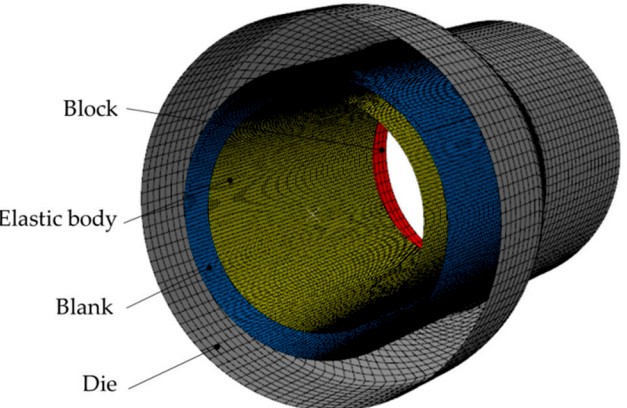

**Figure 7.** Finite element model.

As for an isotropic and incompressible material, the strain–energy function $W$ can be described as a function of principal elongations [20]:

$$W = W(\lambda_1, \lambda_2, \lambda_1^{-1}\lambda_2^{-1}) \tag{1}$$

In the Mooney–Rivlin model, $W$ is described as a function of strain invariants ($I_1$, $I_2$, $I_3$) [21]. The relationship between the principal elongations and strain invariants can be written as:

$$
\begin{aligned}
I_1 &= \lambda_1^2 + \lambda_2^2 + \lambda_3^2 \\
I_2 &= (\lambda_1\lambda_2)^2 + (\lambda_2\lambda_3)^2 + (\lambda_3\lambda_1)^2 \\
I_3 &= (\lambda_1\lambda_2\lambda_3)^2
\end{aligned}
\tag{2}
$$

In addition, $W$ only depends on the first two invariants $I_1$ and $I_2$ because the third invariant $I_3$ is constant which equals one for incompressible materials [22]. When the level of deformation of an

elastic body is relatively low, it is reasonable to apply the original first-order Mooney–Rivlin model, which is expressed as:

$$W = C_{10}(I_1 - 3) + C_{01}(I_2 - 3) \tag{3}$$

where $C_{10}$ and $C_{01}$ represent the material constant of the deviatoric component. In this research, the two coefficients were set at 0.4920 and 0.1752 MPa according to the tensile test result of the rubber material used.

All the meshes of the body were shell elements. The element size of the blank and the elastic body was set at 2.0 mm. The total numbers of elements of the two deformed bodies were 18,960 and 11,280. The friction coefficients were selected as 0.10, 0.10 and 0.18 at the die-blank, blank-blank and elastic body-blank interface, respectively.

### 2.4. Boundary Conditions

To investigate the deformation process in hydroforming process, the maximum forming pressure should be determined, which can be calculated as follows:

$$p_{\max} = \frac{2t}{d}\sigma_b \tag{4}$$

where $t$ and $d$ are the initial wall thickness and the diameter of the overlapping tubular blank, respectively. $\sigma_b$ is the ultimate strength, which can be found in Table 1. The maximum forming pressure was 9.03 MPa obtained from Equation (4). In this research, the forming pressure of 9.0 MPa was selected as the maximum forming pressure.

In addition, wrinkling defects at the overlap are likely to occur because of the material flow along the axial direction at each end of the overlapping blank. Therefore, three boundary conditions, free, tied, axial constraint ends, were selected to investigate the effect of boundary constraints on wrinkling defects. For the tied ends, the outer and inner layers were bound together at each end of the blank. The displacements of the inner and outer layers are equal, as shown in Figure 8a. For the axial constraint ends, the material flow along the axial direction was completely restrained at each end. The displacements of the inner and outer layers are equal to zero, as shown in Figure 8b.

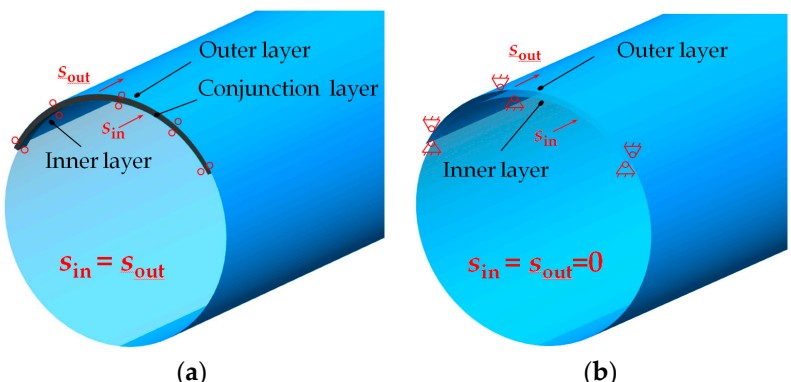

**Figure 8.** Schematic diagram of boundary conditions: (**a**) tied ends; (**b**) axial constraint ends.

## 3. Results

### 3.1. Effect of Boundary Conditions on Wrinkling Defects

Figure 9 shows the deformation process of hydroforming of overlapping blanks with free ends. The blank buckled at the overlap edge of the outer layer at the beginning, as shown in Figure 9a. The wrinkling defects occurred at the outer and inner layer of the overlap as the forming pressure increased, as shown in Figure 9b,c. The material at each end flowed towards the middle area along the axial

direction. Meanwhile, the displacement along the axial direction at the edge was larger than that at the interior of the wrinkling zone. The unequal displacement aggregated the wrinkling defects.

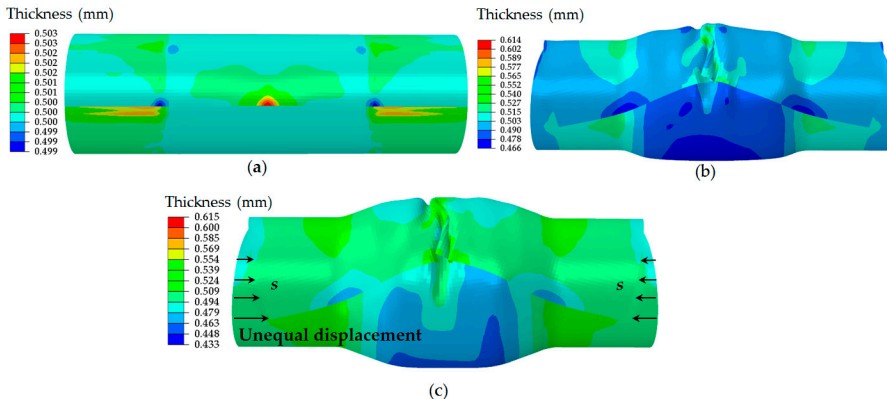

**Figure 9.** Hydroforming lapping blank with free ends (**a**) *p* = 2.0 MPa; (**b**) *p* = 5.0 MPa; (**c**) *p* = 9.0 MPa.

To decrease the difference in the axial displacement and control the wrinkling defects, the overlaps of the blank were tied together at each end. However, slight wrinkling at the outer layer still existed when the ends were tied, as shown in Figure 10. It is indicated that the occurrence of wrinkling defects cannot be avoided if the axial movement of the material at each end is not restricted.

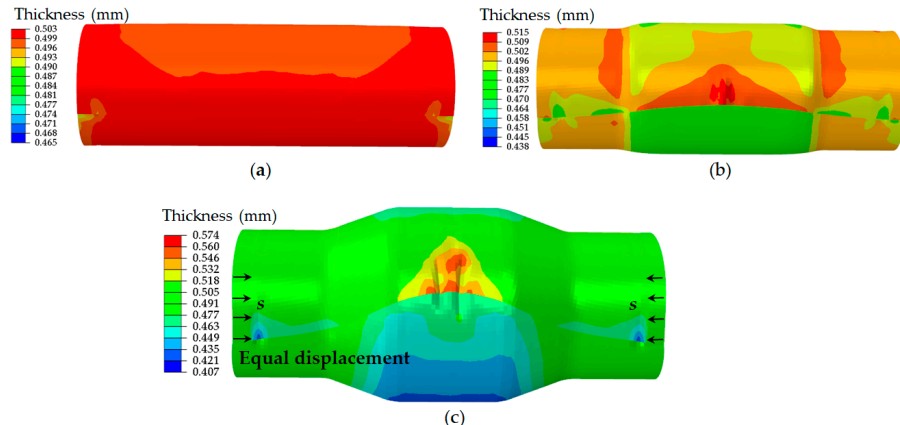

**Figure 10.** Hydroforming of overlapping blank with tied ends (**a**) *p* = 2.0 MPa; (**b**) *p* = 5.0 MPa; (**c**) *p* = 9.0 MPa.

It is indicated that the occurrence of wrinkling defects cannot be avoided if the axial movement of the material at each end is not restricted. Therefore, a blank with axial constraint ends was used to control the wrinkling defects by restricting the flow of the material towards the middle region. The level of wrinkling defects was eased further, as shown in Figure 11. Hence, the axial constraints had a beneficial effect on the hydroforming of overlapping tubular blanks.

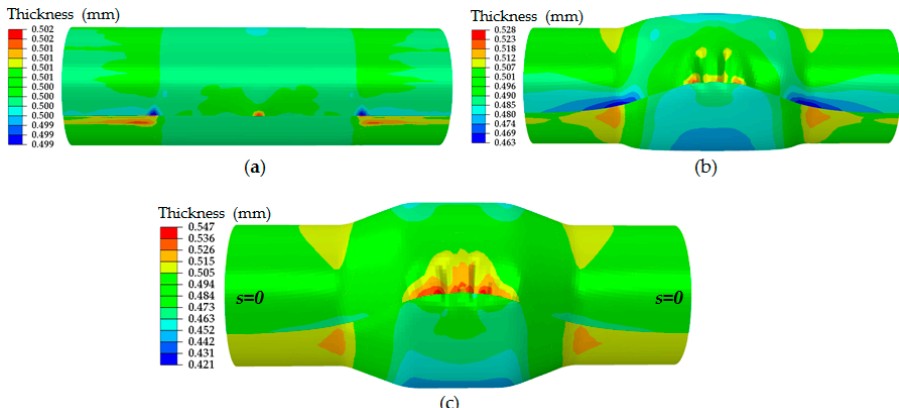

**Figure 11.** Hydroforming of overlapping blank with axial constraint ends (**a**) *p* = 2.0 MPa; (**b**) *p* = 5.0 MPa; (**c**) *p* = 9.0 MPa.

### 3.2. Material Flow Along the Circumferential Direction

The most important difference between the overlapping tubular blank and the tube involves whether the material can flow along the circumferential direction during the hydroforming process. Figure 12 shows the distribution of the circumferential material flow at the edge of the inner layer when the initial overlap angle is 120°. The expansion rates were equal at the middle area of the hydroformed part because the shape of the variable-diameter part was cylindrical. Therefore, the circumferential material flow, or the circumferential supplement, could be reflected in the profile of the blank edge directly. The profile of the edge of the inner layer was concave. That the maximum value was located at the middle cross-section could indicate that the maximum level of the circumferential material flow occurred at the middle cross-section for a plane-symmetric part. Additionally, it could be inferred that the nonuniform circumferential material flow led to a plane-bending deformation.

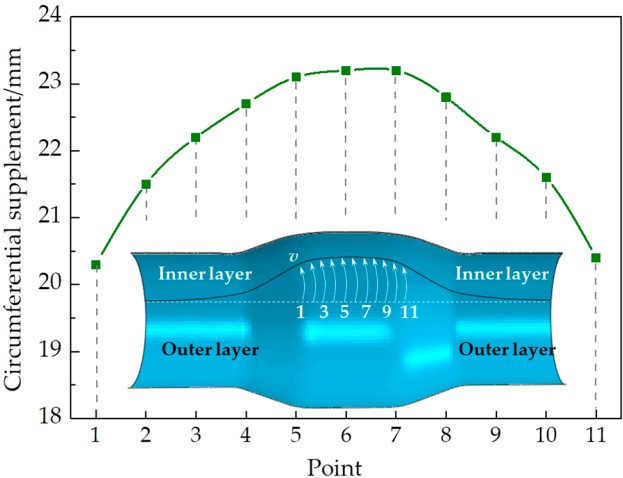

**Figure 12.** Distribution of circumferential material flow at the edge of the inner layer.

### 3.3. Thickness Distribution

To achieve the axial constraints, the arc rings were welded onto the ends of the overlapping blank, as shown in Figure 13a. The perimeter of the blank was 320 mm and the initial overlap angle was 120°. Figure 13b shows the hydroformed part under the forming pressure of 9.0 MPa. The wrinkling defects were scattered at the outer layer of the blank and the wrinkling defects of the inner layer were avoided.

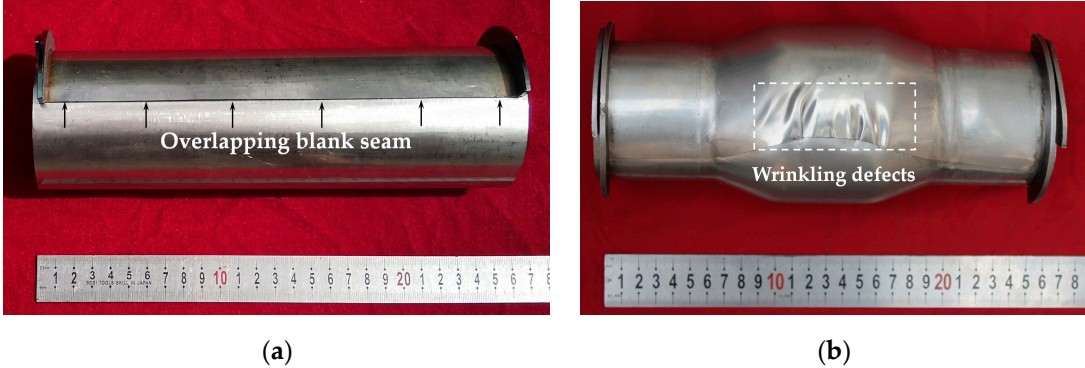

(a)　　　　　　　　　　　　　(b)

**Figure 13.** Hydroforming of overlapping blank under axial constraints in experiment results. (**a**) Initial blank; (**b**) hydroformed part.

Then the redundant overlap and the axial constraints were removed. Then the reserved part was hydroformed again in order to smooth its surface. Finally, a sound part without wrinkling defects was manufactured. Figure 14 shows the final variable-diameter part.

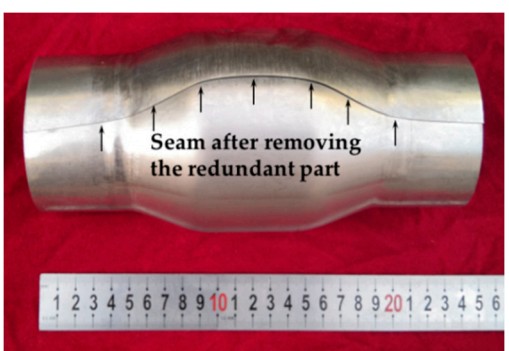

**Figure 14.** The final variable-diameter part.

The thinning ratio distribution was obtained at typical sections of the final part. Figure 15a shows the thinning ratio distribution along the axial direction. The difference in thinning ratio distribution was obvious between the overlap and the opposite side to the overlap. The maximum thinning ratio was 15.9% opposite to the overlap at the bulging region (Point 11) whereas it was only 4.4% at the transition region of the overlap (Point 8). It is indicated that the thickness was nonuniform at the cross-section of the overlapping blank. Detailed variation of the thickness at the middle cross-section is shown in Figure 15b. There were 25 measuring points marked at the middle cross-section. Points 1 and 25 were at the overlap and Point 13 was opposite to the overlap. It can be seen that the thinning ratio continually increased from the overlap towards the opposite. The maximum thinning ratio was 15.9% close to the opposite side of the overlap (Point 12) whereas the minimum thinning ratio was 2.1% at the outer layer of the overlap (Point 25).

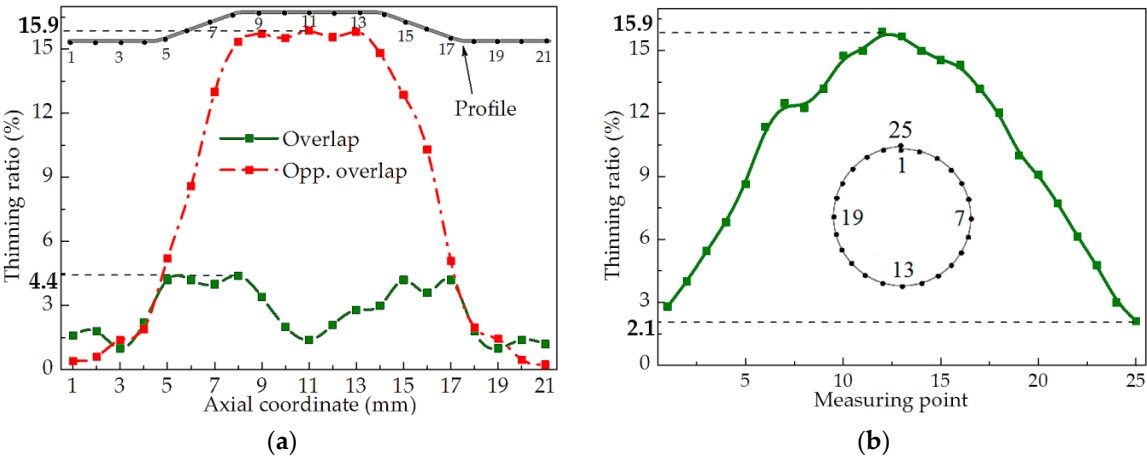

**Figure 15.** Thinning ratio distribution (**a**) along axial direction; (**b**) at middle cross-section.

## 4. Discussion

### 4.1. Stress and Strain Analysis

The simulation and experiment results showed that wrinkling defects are the main forming failure during the hydroforming of overlapping tubular blanks. These wrinkles were usually perpendicular to the axial direction at the overlap. It was implied that the level of compressive axial stress was very high at the region. Figure 16 shows the stress distribution along and opposite to the edge of the outer layer of overlap with both ends of the overlapping blank under axial constraints. The hoop stress and the axial stress increased sharply opposite to the edge, which was analogous to that during the tube hydroforming. On the other hand, the hoop stress was compressive and varied slightly beneath the yield stress and the axial stress was compressive along the edge of the outer layer. Particularly, the compressive axial stress dramatically increased at the axial coordinate of −60 and 60 mm when the internal pressure was 2.0 MPa. With the blank expanding, the maximum compressive axial stress was as many as 589 MPa at the bulging region. Therefore, wrinkling defects occurred at the edge owing to the compressive axial stress during the hydroforming process.

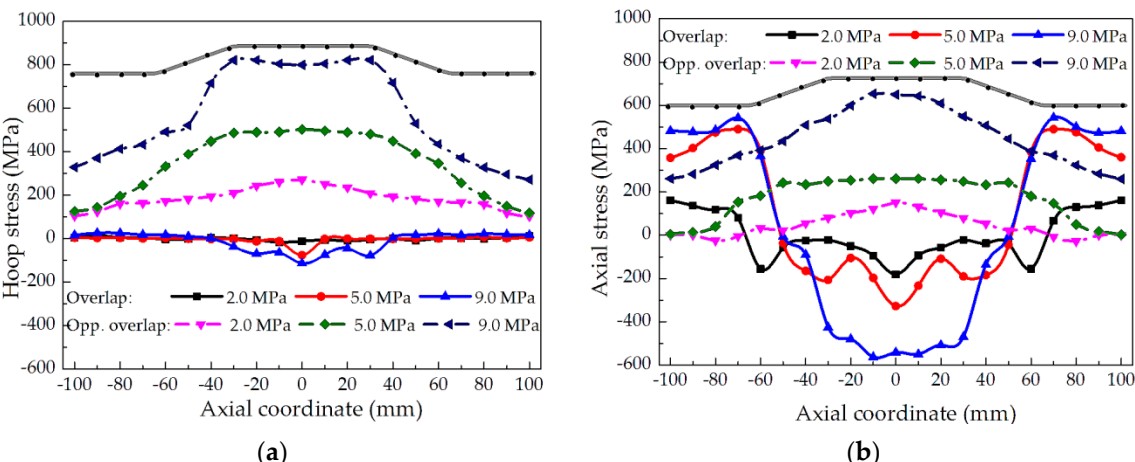

**Figure 16.** Stress distribution along axial direction. (**a**) Hoop stress; (**b**) axial stress.

Although the variation of axial compressive stress was discussed before, the deformation model at the wrinkling region should be determined. Figure 17 shows the axial stress distribution of the outer overlap at the cross-section where the compressive axial stress sharply increases. The increase of compressive axial stress occurred at section A-A when the internal pressure was 2.0 MPa. The axial stress increased from −152 MPa at the edge (Point 1) to 80 MPa at the interior (Point 11). A bending

moment formed at the outer overlap due to the stress distribution at section A-A. This phenomenon was more obvious at section B-B when the internal pressure was 5.0 MPa. The axial stress increased from −325 MPa at the edge (Point 1) to 121 MPa at the interior (Point 11). It is indicated that a bending deformation happens at the edge of the overlap. It is extremely possible that wrinkling defects are caused by a bending moment at the inside radius.

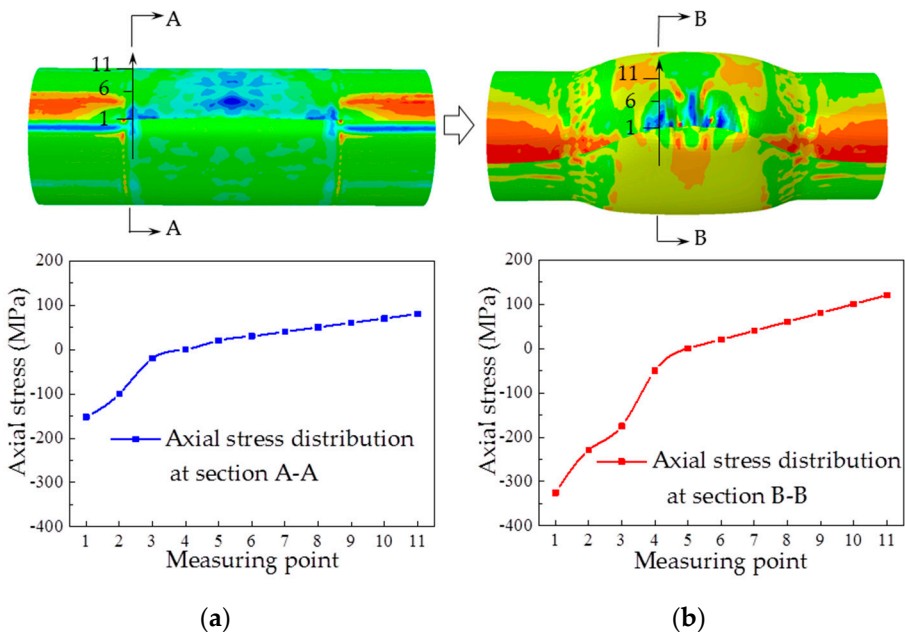

**Figure 17.** Axial stress distribution at the outer layer of overlap. (**a**) *p* = 2.0 MPa; (**b**) *p* = 5.0 MPa.

The strain variation of the hydroformed part was obtained to study the circumferential deformation behavior. Figure 18 shows the circumferential and normal strain distribution at the middle cross-section. The materials of the part elongated along the circumferential direction and thinned along the normal direction in common. The maximum circumferential strain was 0.194 and the minimum normal strain was −0.176 at Point 12. Due to the application of axial constraints, the axial strain of Point 12 was only −0.018, of which the absolute value was much smaller than those of the circumferential and the normal strains. Therefore, the deformation mode was similar to the plane strain state during the hydroforming of an overlapping blank under axial constraints.

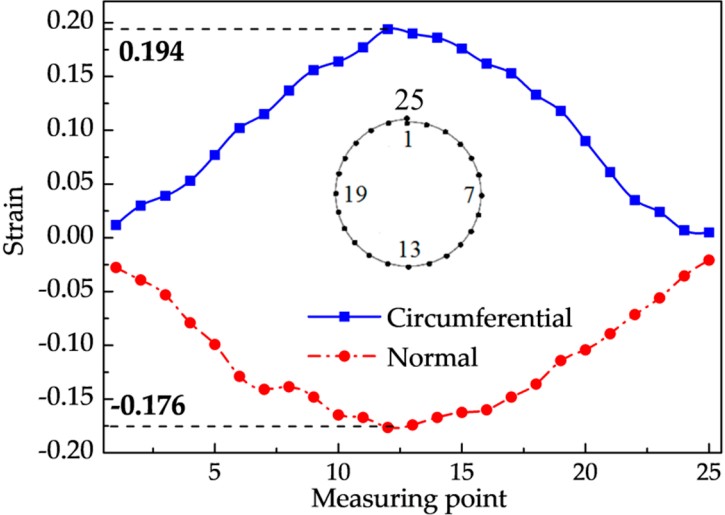

**Figure 18.** Strain distribution at the middle cross-section.

### 4.2. Variation of the Overlap Angle

It is difficult for the overlapping blank to deform simultaneously as its geometry is nonaxisymmetric. Figure 19 shows this deformation process of the overlapping tubular blank when the initial overlap angle $\alpha = 140°$. The deformation sequence was the outer layer of overlap at first, then the inner layer of overlap, and the opposite side to the overlap at last. The deformation had already occurred at the outer layer of the overlap when the internal pressure was 2.0 MPa, as shown in Figure 19b. This value of internal pressure was only about half of the yield pressure, in comparison with hydroforming of tubes with closed cross-sections. The yield pressure in tube hydroforming is usually calculated in Equation (5) [23]:

$$p_y = \frac{2t}{d}\sigma_s \tag{5}$$

where $\sigma_s$ is the yield strength of the material, which can be found in Table 1. Therefore, the yield pressure was 3.78 MPa calculated from Equation (4).

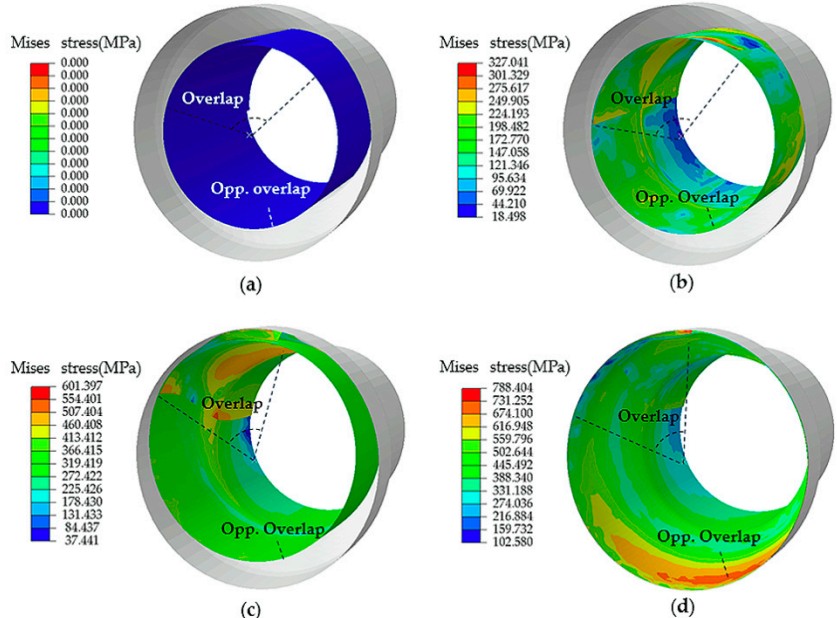

**Figure 19.** Deformation process of the overlapping tubular blank. (**a**) $p = 0$ (initial state); (**b**) $p = 2.0$ MPa; (**c**) $p = 5.0$ MPa; (**d**) $p = 9.0$ MPa.

Figure 19c shows that the entire overlap contacted the die cavity whereas the opposite side to the overlap did not as the internal pressure reached 5.0 MPa, which was only 1.3 times larger than the yield pressure. The bulging height was higher at the overlap than opposite to the overlap at the same level of forming pressure. Finally, the rest of the blank had already contacted the die cavity completely since the internal pressure reached 9.0 MPa, which can be seen from Figure 19d. The majority region was opposite to the overlap where the value of the Mises stress was relatively high. The final overlap angle $\beta$ decreased to 43°.

The circumferential material flow can be also reflected in the variation of the overlap angle, which is defined as follows:

$$\Delta = \alpha - \beta \tag{6}$$

where $\alpha$ and $\beta$ represent the initial and the final overlap angle at the middle cross-section, respectively. Figure 20 shows the variations of the overlap angle under different initial overlap angles. The hydroformed part failed when the initial overlap angle was 80° because the inner and the outer layers were separated. The final overlap angle was negative and its value was −7° in this condition. The value of the final overlap angle was positive when the initial overlap angle was larger than 100°. The

variation of the overlap angle kept increasing as the initial overlap angle increased. It meant that more circumferential supplement of the material was achieved if the initial overlap angle became larger. However, the velocity of increase declined gradually and the variation of the overlap angle was constant since the initial overlap angle was larger than 160°. This indicated that there was a limit of the circumferential supplement during the hydroforming using the overlapping blank.

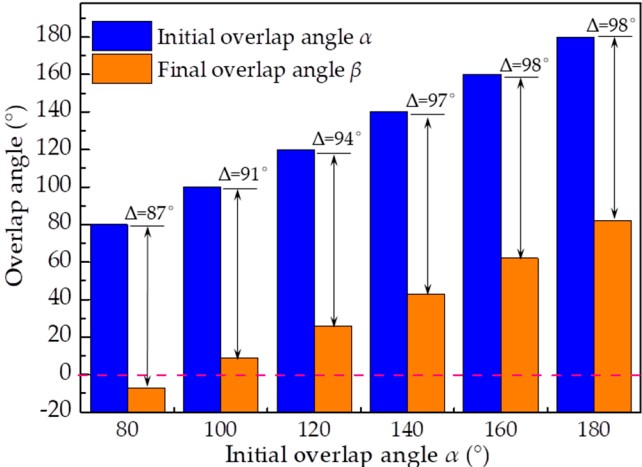

**Figure 20.** The variations of the overlap angle under different initial overlap angles.

### 4.3. Effect of the Overlap Angle on the Thickness Distribution

Figure 21 shows the effect of the overlap angle on the thickness distribution at the middle cross-section when the initial overlap angle ranges from 100° to 180°. The thinning ratios of $\alpha = 100°$ were much smaller than other conditions near the two edges of the hydroformed part. The maximum thinning ratio of $\alpha = 100°$ was larger than that of $\alpha = 120°$. It seemed that the increase of the variation of the overlap $\Delta$ contributed to the improvement of the thickness distribution. However, the maximum thinning ratio became larger with the initial overlap angle increasing since the initial overlap angle was greater than 120°. An optimal thickness distribution was obtained when the initial angle was 120° for the hydroforming of the variable-diameter part with an expansion of 31.6%. To sum up, there was an optimal initial overlap angle to minimize the thinning ratio when taking into account the variation of the overlap angle and the effectiveness of the material flow along the circumferential direction.

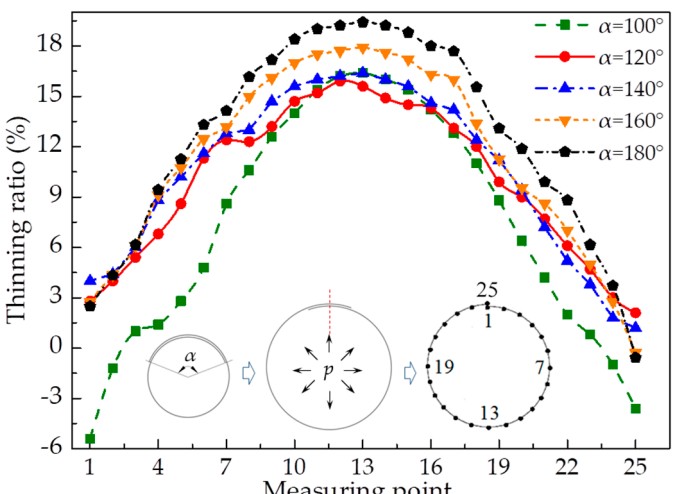

**Figure 21.** Effect of the overlap angle on the thickness distribution at the middle cross-section.

## 5. Conclusions

In this paper, the circumferential material flow was investigated using overlapping blanks with axial constraints in the hydroforming of a variable-diameter part. A self-sealing elastic deformation body was used as a loading tool to store the pressurized liquid. Numerical simulation and experimental methods were used to investigate the deformation features. The detailed results were as follows:

(1) The effect of boundary conditions: free ends, tied ends and axial constraint ends, on wrinkling defects at the overlap was studied. Wrinkling defects can be weakened when the blank with axial constraint ends was used.

(2) Wrinkling defects occurred at the edge of the overlap owing to the compressive axial stress. Furthermore, a bending deformation was observed through the stress analysis.

(3) The deformation of the blank at the overlap initiated at a low pressure that was nearly half of the yield pressure for tube bugling. The contact sequence to the die cavity of the blank was the overlap at first, then the inner layer of overlap, and the opposite side to the overlap at last.

(4) The circumferential material flow was obtained when axial constraints were applied to avoid the influence of the axial material flow. The profile was concave at the edge of the overlap due to the circumferential material flow. The peak value is located at the middle cross-section.

(5) The variation of the overlap angle increased with the initial overlap angle increasing but the improvement of the thickness distribution did not. An optimal thickness distribution was obtained when the initial angle was 120° for the hydroforming of the variable-diameter part with an expansion of 31.6%.

**Author Contributions:** C.H. conceived the experiments and provided all sorts of support during the work; H.F. performed the experiment and wrote the paper. All authors have read and agreed to the published version of the manuscript.

**Funding:** This work was financially supported by the two National Natural Science Foundations of China (project numbers: 51775136 and U1937205). The authors would like to take this opportunity to express their sincere appreciation.

**Acknowledgments:** The presentation about this paper was given at the 9th International Conference on tube hydroforming and the paper was recommended to submit to the journal "metals".

**Conflicts of Interest:** The authors declare no conflict of interest.

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
