# Peer review of "Circumferential Material Flow in the Hydroforming of Overlapping Blanks"

_metals, doi:10.3390/met10070864_

Round 1

Reviewer 1 Report

Overall comments:

Significant revisions were made to the paper and the results were immediately noticed. This updated version of the paper reads well and only a few errors were noticed and are still present. After the remaining errors are revised, it is recommended that this paper be published. 

Specific comments:

line 111 - can still use clarification with explaining the elastic body? how does this fill and expand outwards? what thickness? maybe show another figure with the elastic body? not sure, but I'm still not visualizing how this forms? show and end view of the rubber? If possible, a picture with a note/label is a very easy way to describe what is going on without writing

line 194 too many materials, should change to something like "...circumferential material flow at the edge of the inner layer..."

Figure 19?? Check and revise the Figure sequence numbers. Fig 19 is after Fig 11 and before Fig 13.

Author Response

Please find the author's response in the attachment.

Reviewer 2 Report

  1. What is the unit of Mooney-Rivlin model's constants? MPa?
  2. It is not clear how to weld such tubes after hydroforming.

Author Response

(The authors gave the same response as above.)

Reviewer 3 Report

Dear the Authors,

This manuscript deals with hydroforming process using the end-open tubular blank.

1) In my opinion, because the initial blank was designed and prepared as "the end-open tubular blank",

it should be considered that the elastic recovery behavior (spring-back) is severe.

Also, the circumferential deformation behavior should be addressed.

However, this reviewer did not find ant evidances on this deformation behavior.

2) In addition, the blank was modeled using "shell" elements, so its numerical calculation time maybe relatively short.

However, in general, the blank materials in the hydroforming process should be modeled using 3D brick elements

(8-node brick elements).

That is, the authors should provide the numerical simulation results based on the application of 8-node brick elements.

Author Response

(The authors gave the same response as above.)

Round 2

Reviewer 3 Report

I have no more comments.

This manuscript is a resubmission of an earlier submission. The following is a list of the peer review reports and author responses from that submission.

Round 1

Reviewer 1 Report

Overall Comments:

This paper is fairly well written and introduces a novel and interesting method for performing tube hydroforming operations. However, there are a few areas that this paper needs to improve on before a recommendation for publication can be made. First, it is difficult to follow the authors with their process description. Better or further process explanation and clarification early in the paper is required. The addition of actual process/experimental pictures are suggested at the beginning of the paper as a method to assist with the explanation. Also, this paper needs a thorough proofread for grammatical errors. There were times when the proper tense or missing/improper articles are used. See specific comments for examples of where improvements can be made. Keep in mind that these items are not necessarily an all inclusive list. Further reading is required to

Specific Comments:

Line 26 – “demand of lightweight has been increasing” – lightweight “what” has been increasing? Also, do the authors mean “demand for lightweight…” instead of “demand of lightweight…” – Example of article usage?

Line 28 – “assembled in automotives.” – Do the authors mean, automobiles? Or the automobile industry? – Example of word tense usage?

Line 48 – “axial feeding in the situation.” – Change the to this.

Line 54 – It is unclear what is meant by a self-sealing elastic body? Possibly show a pic?

Line 76/77 – mechanical properties – Were these experimentally determined or from a book? If experimentally determined, how were they determined? Using a sheet or by sectioning a tube section? More clarification needed.

Figure 3 – should show that the True Strain has units (mm/mm).

Line 82 – “In hydroforming process,” – Should be “In the hydroforming process,” – Missing article.

Line 82 – “the blank is open” – What does this mean? Show a pic or better describe/explain what is meant here or earlier in the text.

Line 87 – “The material of blank…” should be “The material blank…” -need to remove “of”.

Figure 5 – This figure is a good idea, but it’s difficult to get a complete visualization to fully understand how the overlapping blanks look like and actually work. The ¼ section removed may actually hinder what is being shown. It is recommended that an exploded isometric figure be inserted to show how the items fit together. The elastic body is labeled, but an actual picture may be better to show what this really is?

Line 126 – “two deformed body” should read “bodies” – Example of where the proper tense needs to be captured and corrected. Again, a thorough proofread is required to catch other instances similar to this.

Lines 152-156 – This explanation is difficult to follow with all of the overlap discussion. Clarification is required.

Line 156 – Explanation of what is meant by closed or open cross-sections is needed. A picture or figure may help with this?

Eqn. 4 and Eqn. 5 – These eqn.s are the same with the exception of the subscripts. Also, Ps is not typically Py (yield pressure).

Line 162 – Figure 8c “shows the entire overlap” – this is not obvious!

Line 164 – “using overlapping blanks” – still not clear about overlapping blanks!

Figure 12 – Actual experimental pictures are always helpful! However, labeling is required to show the overlapping blank seam. This may also help with the open vs. closed cross-section? SHOW MORE PICTURES!

Lines 209-219 – Another difficult to follow paragraph?

Figure 13 – What is the gray line? LABEL!

Line 233 – “increased somewhere” – Using somewhere in a journal paper is never recommended! Use better/different words.

Line 250 – “divided into thinning (dEt<0) and thickening (dEt<0)” – These should have different < or >.

Line 295 – “loading tool to storage the pressurized liquid” – Change “storage” to “store”.

Reviewer 2 Report

Recently, the authors published a very similar study below with only very minor differences with the present work (Abstract and conclusion are almost the same with some minor differences. There is no novelty in current manuscript.):

Cong Han, Hao Feng (2020) A new method for hydroforming of thin-walled spherical parts using overlapping tubular blanks, The International Journal of Advanced Manufacturing Technology, 106, pp. 1543-1552.

The above paper was submitted on 26 May 2019, Accepted 20 November 2019, and published online 19 December 2019.

They referenced some less relevant papers by other authors published in the above journal but avoided referencing the above work. WHY??

The current Metals submission was submitted on 23 December 2019, well after the above paper was online published.

If the authors purposely avoided referencing the paper in the current submission to give the latter an appearance of original work, this is unethical and unacceptable.

The authors should provide sound reasons for this. If sound and convincing reasons are not presented, the manuscript should be rejected.